

# Numerical study of valence band states evolution in $Al_xGa_{1-x}As$ [111] QDs systems

Mikhail Lazarev

Faculty of Computer Science, Laboratory of Methods for Big Data Analysis, HSE University, Moscow, Russia

## ABSTRACT

Quantum dots (QDs) are very attractive nanostructures from an application point of view due to their unique optical properties. Optical properties and valence band (VB) state character was numerically investigated with respect to the effects of nanostructure geometry and composition. Numerical simulation was carried out using the Luttinger–Kohn model adapted to the particular case of QDs in inverted pyramids. We present the source code of the 4-band Luttinger–Kohn model that can be used to model AlGaAs or InGaAs nanostructures. The work focuses on the optical properties of GaAs/AlGaAs [111] QDs and quantum dot molecules (QDMs). We examine the dependence of Ground State (GS) optical properties on the structural parameters and predict optimal parameters of the QD/QDM systems to achieve dynamic control of GS polarization by an applied electric field.

## INTRODUCTION

Low-dimensional nanostructures have unique optical properties influenced by geometry and composition. These properties have led to their diverse applications, ranging from photovoltaics (*Barnham et al., 2002*), light-emitting diodes (LEDs) (*Nakamura et al., 1995*), to laser technologies (*Alferov, 2001*), and so on (*Levine, 1993*). Dimensionality plays a curtail role in light-matter interaction and the behavior of charge carriers, thereby influencing their optical characteristics. In quantum wells (QWs), light emission happens due to the recombination of electrons with heavy holes (*Rau et al., 2000*) in contrast to quantum wires (QWRs), where recombination involves both heavy and light holes (*Vouilloz et al., 1997*). Thus, nanostructure dimensionality impacts on emitted light polarization. QDs have a complex spectral structure where carrier states can be heavy and light with high energy splitting. Their ability to emit single photons positions them as potential integral components in the emerging field of integrated quantum photonic circuits (*Davanco et al., 2017*; *Dietrich et al., 2016*; *Söllner et al., 2015*).

Previous studies have established that the character of holes (heavy or light) in the valence band (VB) of various nanostructures can be modulated by the structural design and material composition of QDs (*He et al., 2007*; *Wang et al., 2008*; *Zhu et al., 2007*).

Corresponding author
Mikhail Lazarev, mvlazarev@hse.ru

Additionally, several methodologies have been identified for achieving dynamic control over the optical properties of QDs. These include the application of external electromagnetic fields (*Fry et al., 2000*; *Bennett et al., 2010*) and strain (*Martín Sánchez et al., 2018*). Several groups have demonstrated electrically driven position control of QD in photonic crystals (*Carter et al., 2013*) and cavity and polarization control in micropillar cavities (*Rakher et al., 2008*; *Reitzenstein & Forchel, 2010*). Further, the modification of emission wavelengths through the application of external stress has been successfully implemented (*Zhang et al., 2014*). The study of dynamic control over QD band structure is fundamental for novel functional materials and devices in a wide range of applications.

QD band structure can be calculated using multiple methodologies, ranging from ab initio approaches (*Reimann & Manninen, 2002*; *Kohn & Sham, 1965*) to empirical pseudopotential method (*Mäder & Zunger, 1994*) and (*Boykin et al., 2007*). While these methods offer detailed insights, their extensive computational demands often render them impractical for simulating heterostructure electronic structures. On the other hand, the **kp** methods are widely applied to describe III–V semiconductor heterostructures optical properties. Application of 4-band and 6-band Luttinger–Kohn **kp** models for (111)-oriented systems were introduced in *Kelsall, Taylor & Wood (1989)*, *Majewski, Stadele & Vogi (1997)*, *Marquardt, O'Reilly & Schulz (2014)*, *Kajikawa (1999)*. There are even higher dimensional models, 8-band (*Tomić, Sunderland & Bush, 2006*; *Andrzejewski et al., 2010*), 10-band (*Gladysiewicz et al., 2013*), and even 30-band models (*Song et al., 2020*), although they lack publicly available source code for verification. A larger model gives more accurate predictions, allowing us to consider more accurate effects like a decrease of optical gain for InGaAs/GaAs QWs with an increase in the hydrostatic pressure (*Gladysiewicz, Kudrawiec & Wartak, 2014*) (10-band *vs* 8-band). However, the needed accuracy depends on the application and experimental setup. Here, to describe nanostructure VB properties, we use a relatively simple 4-band **kp** model implemented in the code (*Lazarev & Karlsson, 2022*), which agrees with experimental data obtained in previous studies. The 4-band **kp** model is a fast solution that describes the VB structure well. The detailed model description is presented in Section 'Method' of this article. Here, we theoretically study the transition process between Light and Heavy holes like VB ground state (GS) characters in QDs of different geometries. We use the same modeling approach as in *Zhu et al. (2009)*, *Troncale et al. (2007)*, *Lazarev, Rudra & Kapon (2023)*.

In this study, we focus on quantum dots (QDs) grown within inverted pyramid structures, as previously explored (*Karlsson et al., 2006*; *Zhu et al., 2010*). Nanostructures in inverted pyramids are well studied experimentally and demonstrate the ability to control the structure composition and geometry with high precision during material deposition. The MOVPE growth technique allows the growing of all possible varieties of nanostructures by controlling heterostructure composition along the growth direction. The schematic illustrations are presented in Fig. 1. Each nanostructure in Fig. 1 represents a nanostructure type that has been experimentally fabricated and reported in the literature; for example QWRs (Fig. 1A) (*Zhu et al., 2006*), QDs (Fig. 1B) (*Zhu et al., 2007*), tailored QDs (*Lazarev, Rudra & Kapon, 2023*; *Szeszko et al., 2012*; *Zhu et al., 2006*) (Fig. 1E), QDMs (*Zhu et al., 2009*) (Fig. 1D) Dot-in-Dot (DiD) (*Troncale et al., 2007*) (Fig. 1C). Different

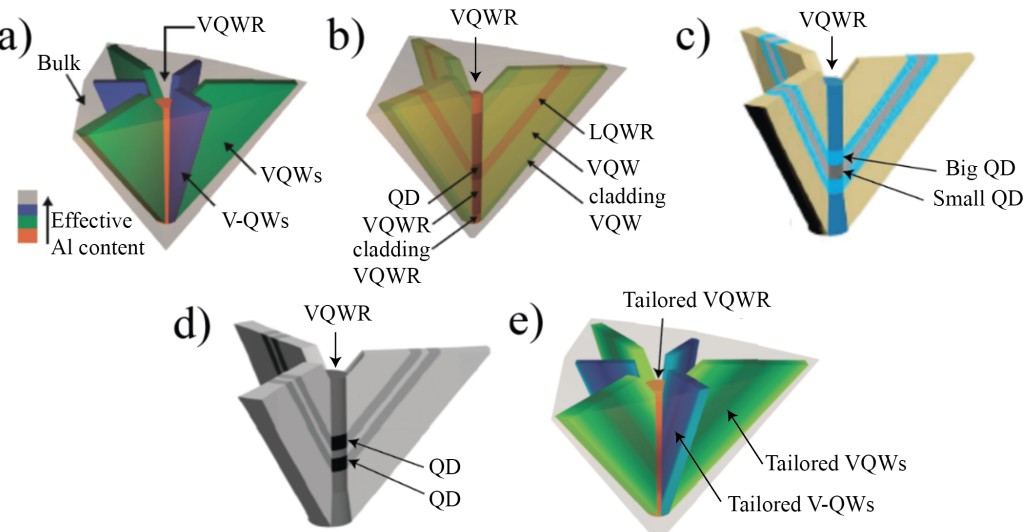

**Figure 1** **Schematic illustration of different kinds of AlGaAs nanostructures embedded in inverted pyramids.** (A) "Long" quantum wire (QWR) laterally bounded by "vertical" QWs, (B) quantum dot, and (C) QD-in-QD heterostructure. (D) QDM of two coupled QDs. (E) Tailored potential QWR/QD. The color codes scale the Al mole fraction in the different parties of the heterostructures.

colors correspond to Al content in the structure in Fig. 1A, schematically presented QWR (symmetry center) in an inverted pyramid. Due to the physical processes of nanostructure growth, QWR formation is always accompanied by the formation of side nanostructures QWs. In Fig. 1A, such side nanostructures are colored green for Vertical QW and blue for Vicinal QWs colors. Summarily to QWR, the QD also forms its side nanostructures, Lateral QWRs. Thus, QD formation in QWR is presented in Fig. 1B. Tailoring of Al content during the growth allows to realize all possible nanostructures in solid form. In this study, we consider the central QWR region. In earlier work (*Zhu et al., 2007*), pyramidal $Al_xGa_{1-x}$ As QDs with heights of 10–100 nm were realized, and as it was reported by changing the QD size, it's possible to control polarization properties. In *Zhu et al. (2007)*, little attention was paid to the related VB states structure.

Achievement of dynamic control over light polarization emitted by the nanostructures in the inverted pyramid is an area of significant interest. Dynamic control of the GS emission polarization of quantum Dot in Dot (DiD) pyramidal nanostructure was proposed in *Troncale, Karlsson & Kapon (2010)*. However, the study used geometrical assumptions of the nanostructure that do not match experimental results, and the source code is not available. In *Troncale (2010)*, polarization optical properties of DiD in a magnetic field study were reported, but polarization switching requires high magnetic field intensities and hence is not feasible for applications.

In this article, we theoretically study VB GSs evolution in pyramidal QD/QDM systems from the geometry and material composition that forms the potential profile. The article is divided into a few parts; in 'Method' introduces the simulation method ('Theory') and outlines the structural design approximations employed ('Modeling of structure

composition'). 'Results and discussion' is dedicated to simulation results; in 'LH and HH states evolution in QD', we present single QD simulation results of GS VB evolution from its length and electric field. In 'QDM Polarization control by barrier adjustment', we study the evolution of the QDM system. In 'Conclusion', we conclude your findings. The Appendix contains the $4 \times 4$ Luttinger Hamiltonian derivation used in this study.

## METHOD

### Theory

Here, we introduce the model with assumptions we used and implemented in the code (*Lazarev & Karlsson, 2022*). Portions of this text here and below were previously published as part of a thesis (*Lazarev, 2019*). As a basis, we start with the Kane Hamiltonian (*Kane, 1957*): $H = H^{k \cdot p} + H_{SO}$, where $H^{k \cdot p} = \left[ \frac{p^2}{2m_0} + V(r) \right] + \left[ \frac{\hbar}{m_0} k \cdot p + \frac{\hbar^2 k^2}{2m_0} \right]$ and $H_{SO} = \frac{\hbar}{4m_0^2 c^2} \sigma \cdot \left[ \nabla V \times p \right]$ thus, the Schrödinger equation looks like:

$$\left( \left[ \frac{p^2}{2m_0} + V(r) \right] + \left[ \frac{\hbar}{m_0} k \cdot p + \frac{\hbar^2 k^2}{2m_0} \right] + \frac{\hbar}{4m_0^2 c^2} \sigma \cdot \left[ \nabla V \times (p + \hbar k) \right] \right) u_k^{(n)}(r) = \varepsilon_k^{(n)} u_k^{(n)}(r) \tag{1}$$

Where $\sigma$ are Paul matrixes, $\nabla V$ the gradient of atomic mean field potential, $u_k^{(n)}(r)$ the Bloch waves, and $m_0$ is the electron mass. Our model works with Hamiltonian (1) and solves it in a 4-band (S and P orbitals) approximation. To achieve more accurate results and consider the influence of other bands, the Löwdin perturbation theory (*Löwdin, 1951*) was implied. See the Supplemental Information for more details.

To each pair of electron and hole corresponds exciton energy and definer as follows: $E_X = E_g + E_{cb} + E_{vb} - E_{Coulomb}^{e-h}$ where $E_{cb}$, $E_{vb}$ are the state energies in CB and VB, and $E_{Coulomb}^{e-h}$ is Coulomb interaction energy. These values depend on QD geometry and material composition. Usually, in small QDs, confinement energy is strong compared to the Coulomb interaction. The Coulomb interaction in the case of strong confinement can be considered as a perturbation (*Warburton et al., 1998*). The perturbation to the Hamiltonian is then written as:

$$E_{Coulomb}^{e-h} = \iint \psi_{cb} \psi_{vb} \frac{e^2}{4\pi \varepsilon_0 \varepsilon |r_{cb} - r_{vb}|} \psi_{cb}^* \psi_{vb}^* dr_{cb} dr_{vb} \tag{2}$$

where $\psi_{vb}, \psi_{cb}$ are the valence and conduction band wavefunctions, $r_{cb}, r_{vb}$ the electron and hole positions.

Observed light intensity and polarization dependence proportional to dipole moment matrix element (*Löwdin, 1951*):

$$I_{eh} = \sum_{S_z = \pm \frac{1}{2}} \left| \sum_{J_z = \pm \frac{3}{2}, \pm \frac{1}{2}} \langle \phi_e^{S_z}(r) | \phi_h^{J_z}(r) \rangle \langle u_e^{S_z} | e \cdot p | u_h^{J_z} \rangle \right|^2 \tag{3}$$

where $e$ is polarization, $p$ is the momentum operator, and $u$ is Bloch wavefunction.

The final expression for polarization dependence in the case of orthogonal LH and HH wavefunctions along and perpendicular z direction is:

$$I_z \propto \left( \frac{4}{3} \langle \varphi_e | \varphi_{LH} \rangle^2 \right) I_{xy} \propto \left( \langle \varphi_e | \varphi_{HH} \rangle^2 + \frac{1}{3} \langle \varphi_e | \varphi_{LH} \rangle^2 \right) \tag{4}$$

Spectral measurements often use the degree of linear polarization (DOLP) value. Here we define DOLP as $DOLP = \frac{I_z - I_{xy}}{I_z + I_{xy}}$. Thus $DOLP = -1$ corresponds to transition HH wavefunction (WF) part and $DOLP = 0.6$ LH WF part.

## Modeling of structure composition

In our analysis, the parameters defining material distribution and geometry are derived from previous research (*Zhu et al., 2006*). The design of the structure's special parameters is based on a simplified model of the "effective" Al-content resulting from the capillarity-driven Al-Ga segregation at the sharp wedges of the. The ratio between "effective" and "nominal" Al content:

$$x_{eff} = \frac{x}{x + K(1-x)} \qquad (5)$$

where $x$ is the nominal (bulk) Al concentration. For different arias of the pyramidal nanostructure $K$ parameter is unique, so $K = 8.9$ for the central QWR region and $K = 2.1$ for side QWs [24]. Since the $x_{eff}$ in an Al content, by definition, is associated with the QWR spectral line (*Zhu et al., 2006*), it leads to an overestimation of actual Al by the blue shift of ∼30 meV (or ∼2% Al content). The overestimation is constant in all points of QWR; thus, it does not affect CB or VB structure. This statement is consistent with experimental results. Effective Al content permits calculating the bandgap profile of the corresponding components of the 3D pyramidal heterostructure. In this article, we implement an additional assumption: due to similar Al content to the bulk, we neglect LQWs regions (blue color in Fig. 1A), assuming Al content to be the same as in the bulk. More geometry details and the ratio of material distribution are also well described in previous studies (*Zhu et al., 2006*; *Lazarev, Rudra & Kapon, 2023*). The bandgap dependence over Al content in GaAs is known from the literature (*Vurgaftman, Meyer & Ram-Mohan, 2001*):

$$E_g(Al_x Ga_{1-x} As) = (1-x) E_g(GaAs) + x E_g(AlAs) - x(1-x) C \qquad (6)$$

where C is the bowing parameter, $E_g(GaAs) = 1.519$ eV and $E_g(AlAs) = 3.1$ eV (low-temperature values). According to Eq. (6), each 1% of Al content gives ∼15 meV to the band gap. The band gap offsets ratio between the CB and VB was taken as ∆CB/∆VB ∼67/33 (*Kopf et al., 1992*; *Langbein, Gislason & Hvam, 1996*).

In our computational framework, $Al_x Ga_{1-x} As$ [111] nanostructures within inverted pyramids are the primary focus, although the code is versatile enough to accommodate any 3D structure specified by the user. The option to add *In* content to GaAs and correspondent Hamiltonian is also implemented. It is important to note that $In_x Ga_{1-x} As$ compound is a material with internal strain, and the Hamiltonian presented above must be modified. While our current discussion does not delve into the strain-modified Hamiltonian, this feature is also incorporated into the code. The numerical solution of the Hamiltonian was performed by the finite difference method.

An example of the model structure geometry is presented in Fig. 2, which is a 10 nm thick GaAs QD embedded in AlGaAs barriers. The modeled area is 40 ×40 nm size which is enough to make sure that border conditions (infinite barriers) do not have an effect on the

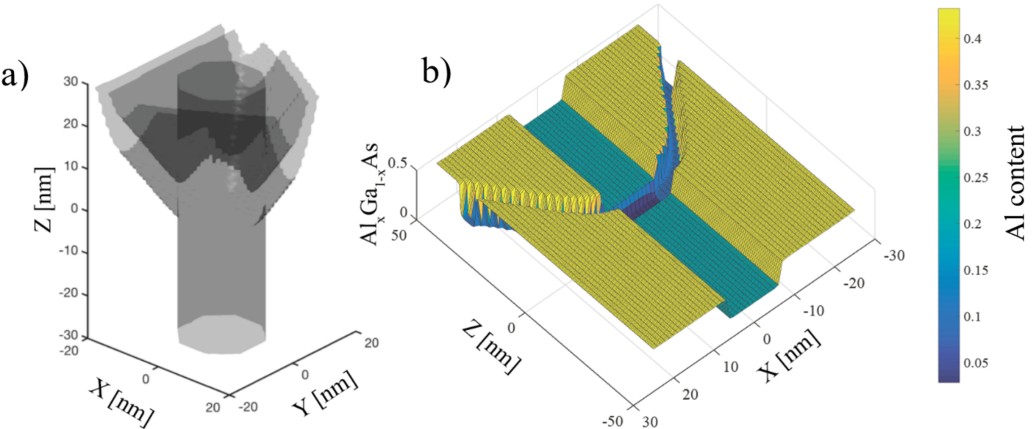

**Figure 2** **The example of the model structure geometry is presented, 10 nm thick GaAs QD embedded in AlGaAs barriers.** (A) 3D isosurface of Al content ($x = 0.4$) in a simulated 10 nm thick GaAs QD nanostructure. (B) Z-X cut of the simulated nanostructure showing Al content distribution.

exciton structure in the area of interest. The calculation grid parameter is chosen as 0.1 nm for all simulation results presented in this article. The diameter of the QD/QWR cylinders is kept at 18 nm. Figure 2B illustrates the Al distribution in the ZX plane. To reiterate key parameters, the effective Al content within the core cylinder of the QD is approximately 2.8%, and in the vertical barrier cylinders, it stands at about ∼7.2% (for 40% nominal Al content).

This methodology was successfully applied to analyze experimental results. *Lazarev, Rudra & Kapon (2023)*, the impact of valence band mixing and spectrum of confined states on the polarization of emitted light from QDs with tailored potential was explored as a function of confinement potential shape that allows it to carefully examine the effect of nanostructure dimensionality on nanostructure emission properties.

## RESULTS AND DISCUSSION

### LH and HH states evolution in QD

Figure 3A presents the calculated energies of the lowest CB and VB (Fig. 3B) states as a function of QD thickness $t$, exclusive of Coulomb interaction effects. The height $t$ of the QD cylinder (the QD "thickness") is varied between 8 to 20 nm (see inset in Fig. 3A). The effective Al content in the core cylinder of the QD is ∼2.7% (for 20% nominal Al content. In the vertical barrier cylinders, it is ∼7.2% (for 40% nominal Al content), which is illustrated in Fig. 3E as a dark and light region in the vertical wire region (pyramid center). The LH (blue color) and HH (red color) states cross in energy for QD thickness $t \sim 13$ (see Fig. 3B). Remarkably, this transition also takes places in the VB mixing characteristic (Fig. 3D), where at the crossover point, the HH and the LH WF portions are the same for the first two VB states. For $t < 13$ nm, the ground state is HH-like, with smaller $t$ the structure behaves in his respect as a "thin" QD. For $t > 13$ nm, the ground state is LH-like, with larger $t$ the structure behaves as a "QWR" or elongated QD. These results

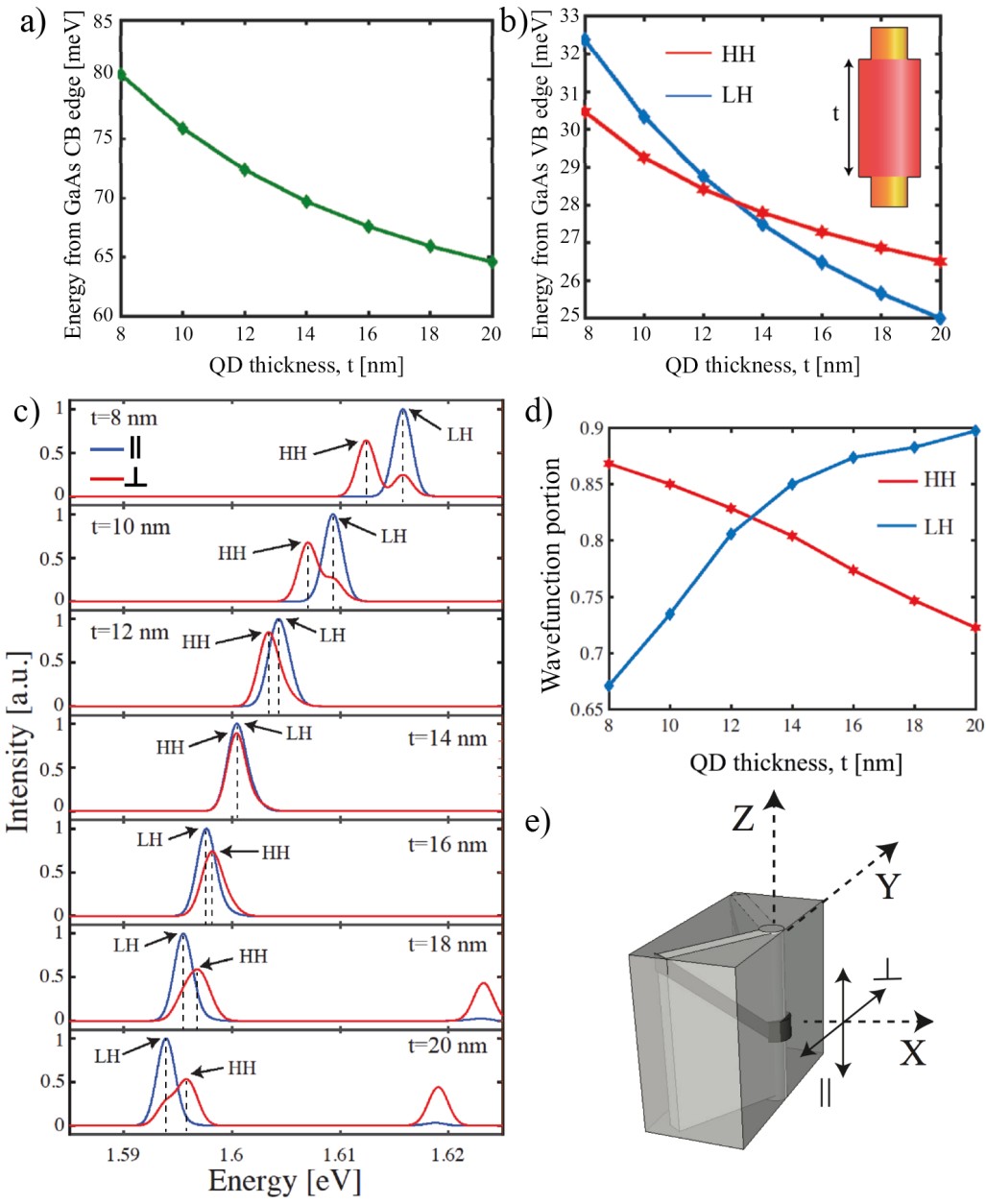

**Figure 3** **GS QD polarization-resolved optical spectra and VB structure for various thicknesses.** (A) Calculated dependence of energy of first CB state from QD thickness. (B) Calculated dependence of energy of first two VB states from QD thickness. Blue color corresponds to LH transitions, and red color to HH transitions. (C) Calculated absorption spectra of $t = 8$ nm, 10 nm, 12 nm, 14 nm, 16 nm, 18 nm, 20 nm QD thickness. Different colors correspond to different polarizations. Blue is emission polarized in the Z direction and red in the XY plane. (D) Calculated wavefunction portions of the first two VB states from QD thickness. The red color is the state with HH, and another one by the blue color with LH major parts. (E) Schematic illustration of QD in an inverted pyramid with the indication of parallel and perpendicular light polarization.

suggest the searching area to choose QD parameters for achieving a structure at the border between HH and LH behavior of the VB ground state. To achieve more accurate results the Coulomb interaction must be considered.

The calculated polarization-resolved optical spectra of this QD structure are shown in Fig. 3C for various thicknesses $t$; red and blue curves correspond to linear polarization normal and parallel to the growth direction. Figure 3E schematically presents light polarization axes with respect to the nanostructure. Line broadening of 2 meV was applied, and Coulomb interaction was taken into account as a perturbation. This set of spectra shows how the GS transition gradually changes its character from LH to HH with a reduction of QD thickness. Interestingly, the switching from HH to LH–like lowest energy state occurs near $t = 14$ nm (as in Fig. 3B), which is different on 1nm from switching simulated without coulomb interaction.

In a range of $t \sim 14$ nm, the QD GS transition changes its polarization with the "switching" of the VB ground state character from LH to HH. Since, around this point, small variations in the confinement potential can yield such GS character "switching", such QD can serve as a base for achieving the desired tunable transition using an electric field. Here, we analyze the effect of an electric field on such "equilibrium": QD structure.

To illustrate the effect of an electric field, we consider the same QD structure discussed above with fixed QD thickness at $t = 14$ nm. The confined CB and VB states are then computed by adding an electric field E oriented in the growth direction z. From the experimental point of view, considering the structure size of L< 1μm the voltage magnitude at $T =10$K that can be applied is about 3V, it gives us a feasible range of electric field simulation up to $E_{max} = 30$ [kV/cm].

Figure 4A shows the calculated optical spectra of the QD structure for an electric field of amplitude $E = 20$ [kV/cm] oriented along the growth direction. We would like to highlight two significant differences compared to the spectrum for $E = 0$. First, the absorption edge is red-shifted due to a quantum confined Stark effect (*Fry et al., 2000*). Secondly, the lowest energy transition line with nearly equal H and V light intensity components develops for $E > 0$ into a richer structure with characteristic polarization features. This is illustrated by DOLP spectrum, which shows an HH-like feature at low energies instead of $DOLP \sim 0$ for $E = 0$. The reason is that the electric field changes the QD potential profile in such a way that the lowest VB state becomes HH-like, and the Stark shifts are different for the different VB states. Thus, applying an electric field can tune the QD structure to change the polarization state of light emitted by the QD ground states. The impact of the electric field on the lowest transition state energy is displayed on the right panel of Fig. 4. The spectral lines near 1.595 eV in Fig. 4 are related to the first CB and first two VB transitions. Note that the polarization is determined by the part (LH or HH) of the VB wavefunction that best overlaps with the CB state.

## QDM polarization control by barrier adjustment

QDM systems contain many parameters that affect QDs coupling strength, and thus, system's optical properties, the main geometrical parameters, are illustrated schematically in Fig. 5. For example, here, we show the effect of barrier height on the polarization

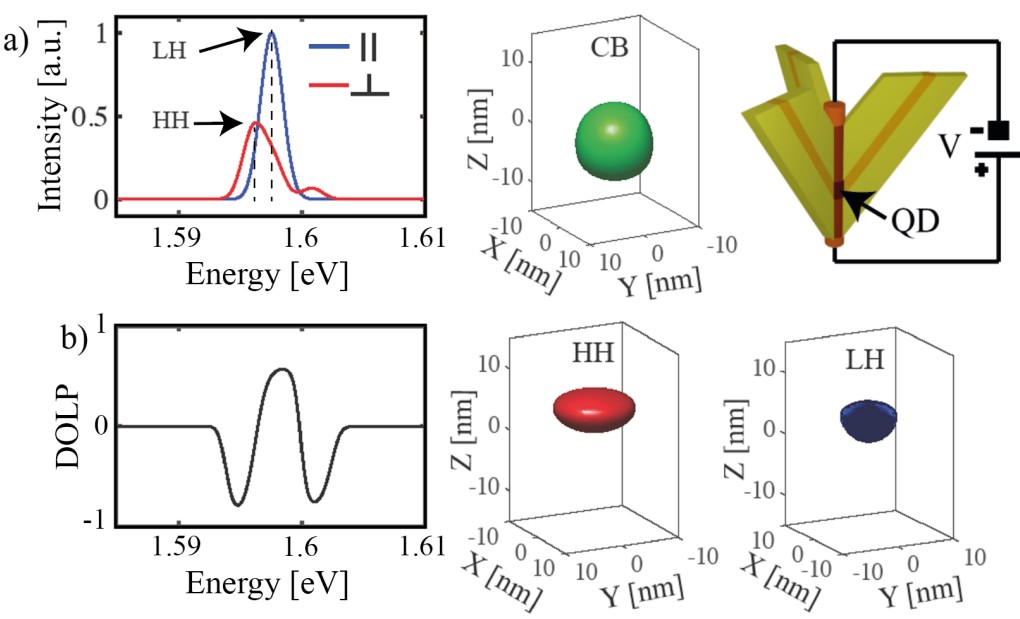

**Figure 4** **QD ($t = 14$ nm), subject to an electric field $E = 20000$ V/cm, applied in the z direction.** Left panel: (A) Calculated optical spectra taking into account Coulomb interaction and 2 meV broadening. (B) DOLP spectra. Right panel: schematic presentation of QD in an inverted pyramid with electric contact and wavefunctions iso-surfaces of the GS CB state and parts of two lowest VB states that overlap with GS CB: green–electron; red–HH; blue-LH.

switching of the QDM GS emission. Figure 6 presents the dependence of the lowest CB and VB states on Al content (in 'nominal' values) in the outer barrier (nominal Al content in QD core: 20%; in external barriers: 40%, QD thickness $t = 7$ nm; barrier thickness $d = 5$ nm). The GS of the CB increases with increasing Al content as the potential energy of the barrier increases. The VB states dependence is more complex: in between the two extreme barrier heights, the VB GS changes its character from LH- to HH- like with increasing Al content (see Fig. 6B).

Figure 7A presents a comprehensive summary of the calculated optical spectra, considering Coulomb interaction as a perturbation and a line broadening of 2 meV. The lowest energy transition is mainly LH-like for low barriers and becomes degenerate with the HH-like one at ~40% Al. The spectra show the evolution of the HH and LH transition lines, their crossing when Al content becomes ~40%, and at 50%, the complete change of the GS to HH one. Figure 7B shows the spectral positions of the GS and the first excited transition *versus* the barrier Al content, indicating the VB character (red color for HH, blue for LH). Figure 7C shows the VB mixing dependence on the barrier height. In summary, the common trend for all structures is that at certain inner QDs coupling regimes (from strong to low), the GS transition changes between LH to HH-like. As a consequence, the emission polarization of the GS also changes. This figure illustrates the two regimes of behavior, with two different GS transition types and polarization of emitted light. The QDM system at the intermediate point where the HH and LH states are almost

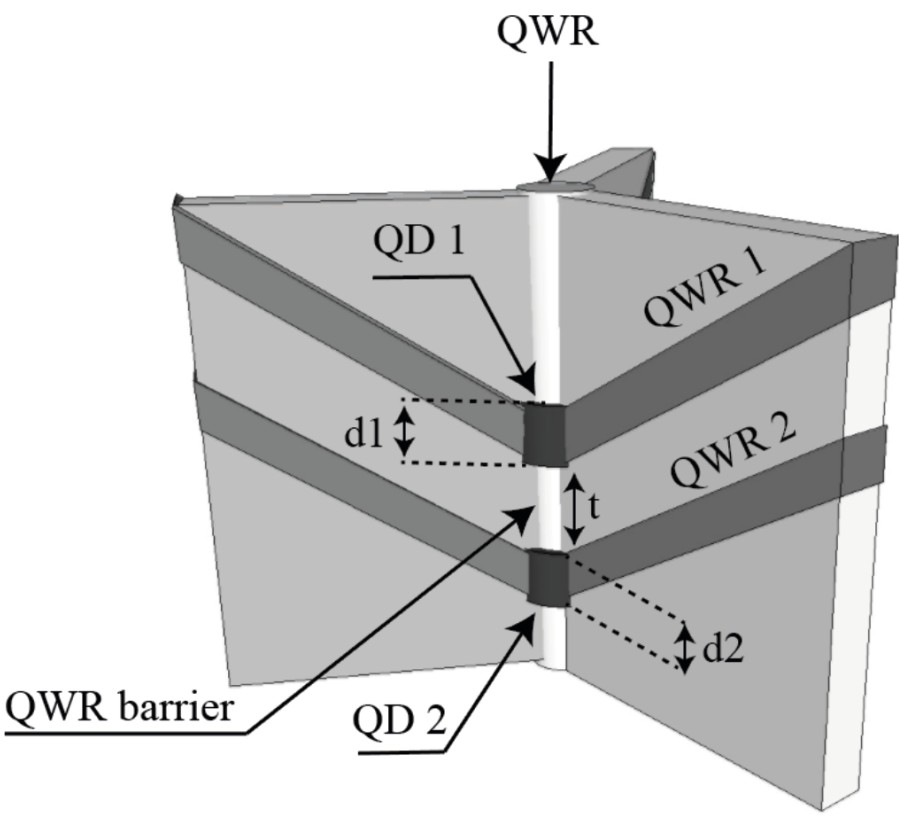

**Figure 5** **Schematic illustration of two QDs system (QD molecule) embedded in an inverted pyramid.**
The color illustrates the Al mole fraction in the different parties of the heterostructures.

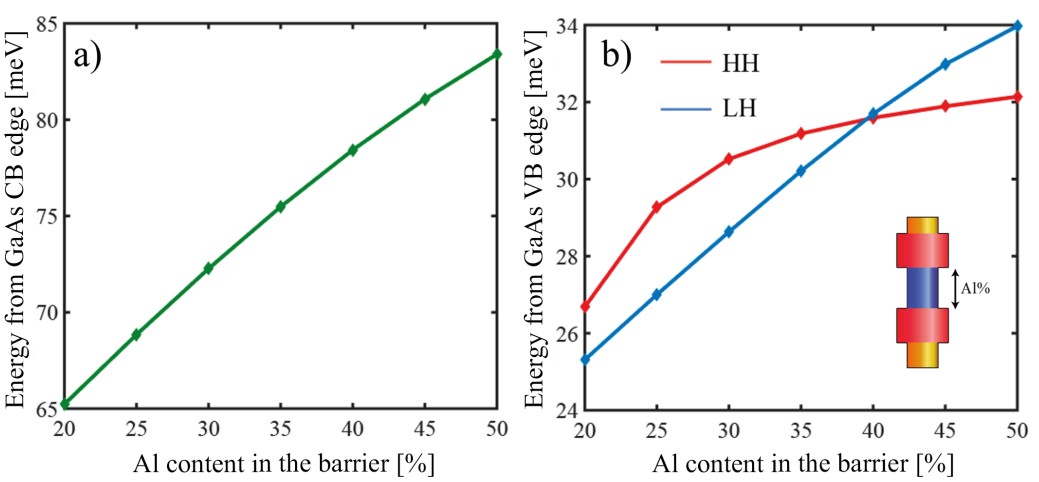

**Figure 6** **QDM CB and VB states control by barrier adjustment.** (A) Calculated energy dependence of
the first CB state *versus* the QDM barrier composition (in 'nominal' values of Al content). (B) Calculated
energy of the first two VB states from QD barrier composition. The blue line corresponds to LH transi-
tion, red to HH *transition*. The inset is a schematic of the QDM core part.

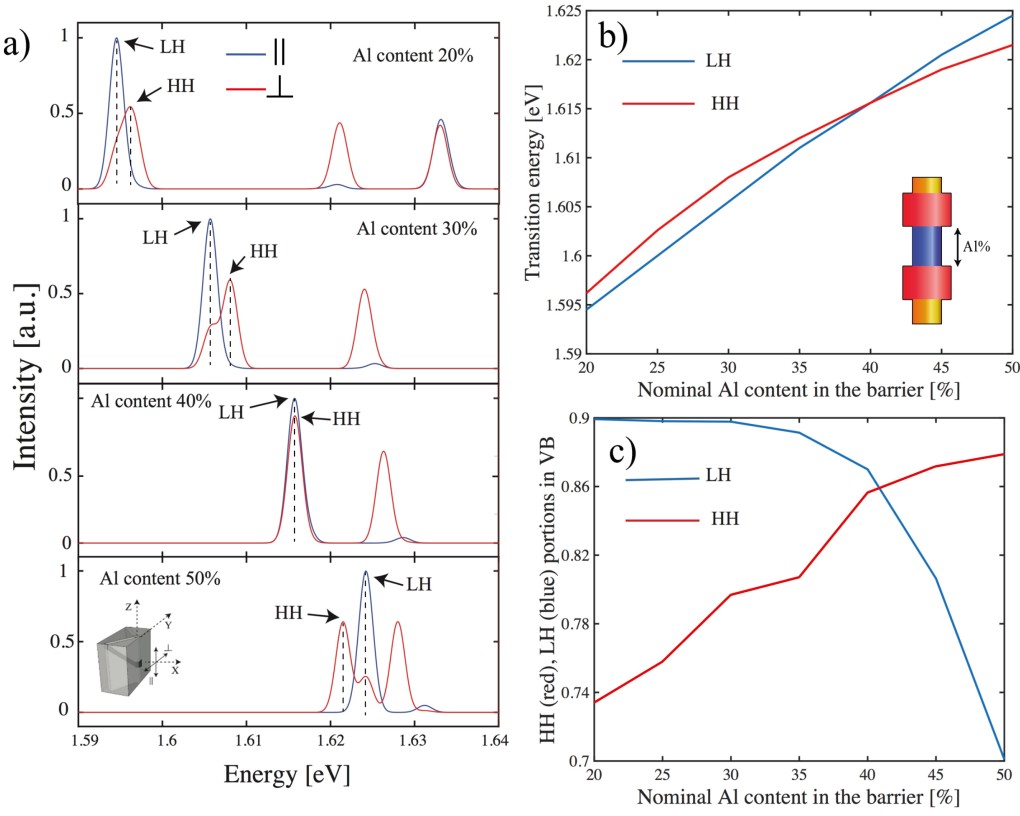

**Figure 7 QDM spectra study dependence from the barrier.** (A) Calculated absorption spectra of the QDM for different heights of the barrier (in 'nominal' values Al content). Lines corresponding to HH-like and LH-like transitions are indicated. (B) Calculated energies of the lowest energy transitions (A) *versus* QDM thickness barrier height. Different colors correspond to different transition types, the red line between CB and HH transition blue line between CB and LH transition. (C) Major wavefunction parts of the first two VB states dependence, HH-red and LH-blue.

degenerate opens an opportunity to tune the GS emission character by applying an external electric field due to the sensitivity of such structure.

Next, we consider the effect of an external electric field in a QDM with the following structural parameters: QD cores of 20% Al content and thickness $t = 7$ nm each, QD external barriers of 40% Al content, QD inner barrier thickness $d = 5$ nm, and 35% Al content. Figure 8 shows the simulation results of the system without an external electric field ($E = 0$). Figure 8A presents the calculated optical spectra, and Fig. 8B the corresponding DOLP spectra. In this configuration, the GS LH and HH transition is almost at the same energy, with a more complex structure of the excited states transitions. The right panel of Fig. 8 shows a side view of the lowest CB and VB states WFs isosurfaces.

For the same QDM structure, the other potential asymmetry induced by an external electric field of $E = 0.9$ [kV/cm] aligned in the growth direction dramatically changes the optical spectra and WFs shapes (Fig. 9). As in the case of the single QD with the applied electric field, the polarization of the GS transition and its energy change. In contrast to

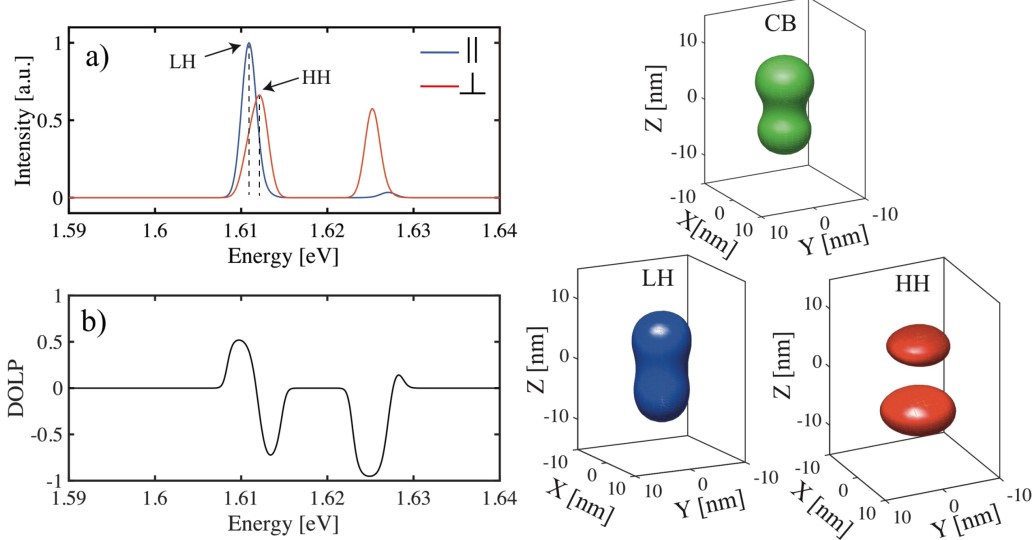

**Figure 8** **QDM structure nominal 35% Al content in the barriers.** QDM structure (see text for parameters) without an electric field. Left panel: (A) Calculated absorption spectra taking into account Coulomb interaction and 2 meV broadening. (B) DOLP spectra. Right panel: wavefunctions iso-surfaces of the GS CB state and parts of two lowest VB states that overlap with GS CB: green–electron; red–HH; blue - LH.

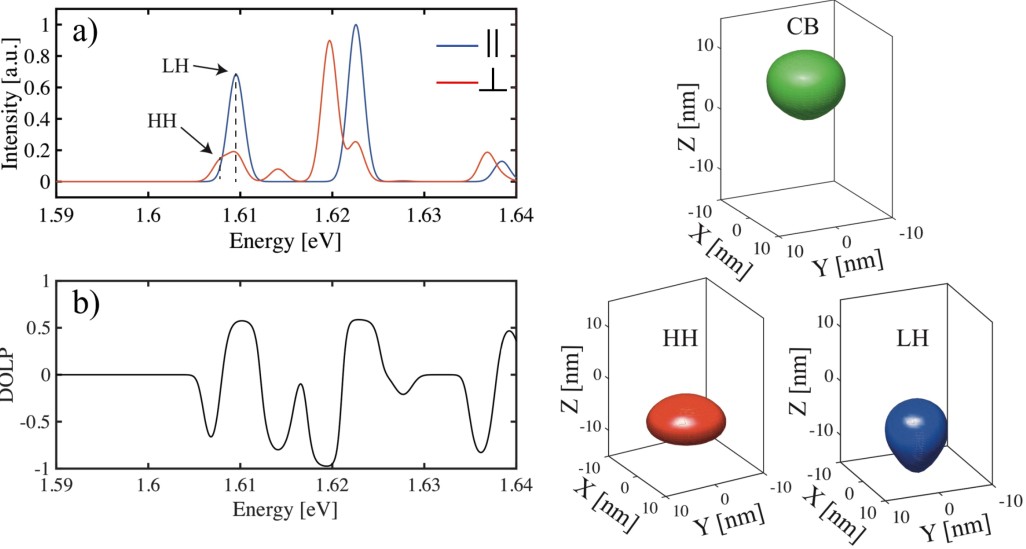

**Figure 9** **DM structure (see text for parameters) subject to an electric field $E = 9000$ V/cm, applied in the +z direction.** Q Left panel: (A) Calculated absorption spectra taking into account Coulomb interaction and 2 meV broadening. (B) DOLP spectra. Right panel: wavefunctions iso-surfaces of the GS CB state and parts of two lowest VB states that overlap with GS CB: green–electron; red–HH; blue-LH.

the case $E = 0$, the lowest energy transition now has $DOLP = -0.6$ instead of $DOLP = 0.5$ to its more important HH character of the VB state involved. As usual, the electric field pushes the CB and VB WFs in the opposite direction for a Stark effect. This also leads to a reduction in the transition strength due to the smaller WF overlaps.

The case of two-QDs molecule can be extended to N QDs (QDSLs), which leads us to the formation of minibands (*Sugaya et al., 2010*), as was also shown in the case of QW superlattices (*Dingle, 1975*). Such systems can also be realized experimentally in inverted pyramids as a vertically aligned coupled QD array. In QDSLs, the barriers height and width are crucial parameters that define the coupling strength between the QDs. In such systems with modulated internal potential in a low coupling regime, GS HH WFs can have as large as LH WF in QDs with flat potential. However, the corresponding emission will have a different polarization. The consequence of low coupling is that a ''forest'' of close-spaced optical transition lines appears. Under the application of an electric field, the excited CB and VB WFs overlap due to tunneling to neighbor QDs, while the first CB and VB states are separate and do not overlap (*Lazarev, 2019*). From an experimental point of view, it's hard to fabricate and analyze the VB structure of N QDs due to structural complexity, optical line merging, and non-linear effects during the nanostructure growth.

## CONCLUSION

The sensitivity of VB state structure to the environment of QD and QDM makes them as promising candidates for polarization-controlled single photon sources. A number of QDs were simulated by 4-band Luttinger–Kohn **kp** model with QDs thickness from 8 nm and 23 nm in length. The ''equilibrium'' point where LH and HH have the same energy positions was identified in QDs approximately 14 nm in length. Applying an electric field along the QD allows us to control the VB structure so that VB GS character becomes HH or LH, which can be observed experimentally by measuring DOLP. Modulating the coupling strength in QDM by the potential variations in the barrier between the QDs we study QDM sensitivity of optical properties to its structural parameters. We show that QDMs are more sensitive to an external electric field than single QDs and demonstrate how it affects QDM absorption spectra. This knowledge will be helpful for the construction of future polarization-switching devices.

The model source code (*Lazarev & Karlsson, 2022*) may be helpful for similar studies or educational purposes. The code is available on GitHub: https://github.com/Mikelazarev/Low-dimensional-band-structure-model-.

## ACKNOWLEDGEMENTS

My acknowledgments go to Prof. Fredrik Karlsson for providing numerical codes. Also, I would like to thank Prof. Eli Kapon for the useful discussions.

## APPENDIX

The Hamiltonian in matrix form, considering SO interaction, looks like this: $H = H_0 + H' + H^{SO}$ where:

**Table 1 Eigenvalues and corresponding eigenstates of the CB and VB states.**

| Eigenvalues and corresponding eigenstates at $k = 0$ | | | |
|---|---|---|---|
| Energy | First Kramers Set | Second Kramers Set | Name |
| $E_c$ | $\left\lvert\frac{1}{2},-\frac{1}{2}\right\rangle = \lvert S\downarrow\rangle$ | $\left\lvert\frac{1}{2},\frac{1}{2}\right\rangle = \lvert S\uparrow\rangle$ | Electron |
| $E_v'$ | $\left\lvert\frac{3}{2},\frac{3}{2}\right\rangle = \frac{i}{\sqrt{2}}\lvert(X+iY)\uparrow\rangle$ | $\left\lvert\frac{3}{2},-\frac{3}{2}\right\rangle = \frac{-i}{\sqrt{2}}\lvert(X-iY)\downarrow\rangle$ | Heavy Hole |
| $E_v'$ | $\left\lvert\frac{3}{2},\frac{1}{2}\right\rangle = \frac{-i}{\sqrt{6}}\lvert(X+iY)\downarrow - 2Z\uparrow\rangle$ | $\left\lvert\frac{3}{2},-\frac{1}{2}\right\rangle = \frac{i}{\sqrt{6}}\lvert(X-iY)\uparrow + 2Z\downarrow\rangle$ | Light Hole |
| $E_v' - \Delta_0$ | $\left\lvert\frac{1}{2},\frac{1}{2}\right\rangle = \frac{-i}{\sqrt{3}}\lvert(X+iY)\downarrow + Z\uparrow\rangle$ | $\left\lvert\frac{1}{2},-\frac{1}{2}\right\rangle = \frac{-i}{\sqrt{3}}\lvert(X-iY)\uparrow - Z\downarrow\rangle$ | SO Hole |

$$H_0 = \begin{pmatrix} H_1 & 0 \\ 0 & H_1 \end{pmatrix} \text{ where } H_1 = \begin{pmatrix} E_c+\varepsilon & iP_0k_x & iP_0k_y & iP_0k_z \\ -iP_0k_x & E_v+\varepsilon & 0 & 0 \\ -iP_0k_y & 0 & E_v+\varepsilon & 0 \\ -iP_0k_z & 0 & 0 & E_v+\varepsilon \end{pmatrix}$$

$$H^{SO} = \frac{\Delta_0}{3} \begin{pmatrix} 0 & 0 & 0 & 0 & 0 & 0 & 0 & 0 \\ 0 & 0 & -i & 0 & 0 & 0 & 0 & 1 \\ 0 & i & 0 & 0 & 0 & 0 & 0 & 0 \\ 0 & 0 & 0 & 0 & 0 & -1 & i & 0 \\ 0 & 0 & 0 & 0 & 0 & 0 & 0 & 0 \\ 0 & 0 & 0 & -1 & 0 & 0 & i & 0 \\ 0 & 0 & 0 & -i & 0 & -i & 0 & 0 \\ 0 & 1 & i & 0 & 0 & 0 & 0 & 0 \end{pmatrix}$$

$$H' = \begin{pmatrix} H_2 & 0 \\ 0 & H_2 \end{pmatrix} \text{ where } H_2 = \begin{pmatrix} 0 & 0 & 0 & 0 \\ 0 & L'k_x^2 + Mk_y^2 + Mk_z^2 & N'k_yk_x & N'k_zk_x \\ 0 & N'k_yk_x & Mk_x^2 + L'k_y^2 + Mk_z^2 & N'k_yk_z \\ 0 & N'k_zk_x & N'k_zk_y & Mk_x^2 + Mk_y^2 + L'k_z^2 \end{pmatrix}$$

$A'$, $B$, $L'$, $M$, $N'$ are second order interactions due to Löwdin renormalization involving states outside the S, P subspace. $\varepsilon = \frac{\hbar^2k^2}{2m_0}$ and $\Delta_0$ is the spin–orbit splitting: $\Delta_0 = \frac{-3i\hbar}{4m_0c^2}\langle X\rvert \frac{\partial V}{\partial x}p_y - \frac{\partial V}{\partial y}p_x \lvert Y\rangle$. In case of GaAs, $B = 0$.

This Hamiltonian contains four atomic bands, kinetic, spin–orbit, perturbation of five next bands, and atomic potential in terms of mean-field. Because the transition rate is proportional to $\langle\varphi_{cb}\rvert e\cdot p\lvert\varphi_{vb}\rangle$ where $P$ is momentum $e$ is a polarization vector. For optical studies, it's more convenient to change the basis of this Hamiltonian to the basis of eigenfunctions of the momentum operator (HH and LH basis). A set of new basis wavefunctions are presented in the following Table 1 (*Davies, 1997*).

New matrix elements in this basis looks like:

$$H_0 = \begin{pmatrix} E_c+\varepsilon & 0 & V & 0 & \sqrt{3}V^* & -\sqrt{2}U & -U & \sqrt{2}V \\ 0 & E_c+\varepsilon & -\sqrt{2}U & -\sqrt{3}V & 0 & -V^* & \sqrt{2}V^* & U \\ V^* & -\sqrt{2}U^* & E_v & 0 & 0 & 0 & 0 & 0 \\ 0 & -\sqrt{3}V^* & 0 & E_v & 0 & 0 & 0 & 0 \\ \sqrt{3}V^* & 0 & 0 & 0 & E_v & 0 & 0 & 0 \\ -\sqrt{2}U^* & -V & 0 & 0 & 0 & E_v & 0 & 0 \\ -U^* & \sqrt{2}V & 0 & 0 & 0 & 0 & E_v & 0 \\ \sqrt{2}V^* & U^* & 0 & 0 & 0 & 0 & 0 & E_v \end{pmatrix}$$

$$H^{SO} = \frac{\Delta_0}{3} \begin{pmatrix} 0 & 0 & 0 & 0 & 0 & 0 & 0 & 0 \\ 0 & 0 & 0 & 0 & 0 & 0 & 0 & 0 \\ 0 & 0 & 1 & 0 & 0 & 0 & 0 & 0 \\ 0 & 0 & 0 & 1 & 0 & 0 & 0 & 0 \\ 0 & 0 & 0 & 0 & 1 & 0 & 0 & 0 \\ 0 & 0 & 0 & 0 & 0 & 1 & 0 & 0 \\ 0 & 0 & 0 & 0 & 0 & 0 & -2 & 0 \\ 0 & 0 & 0 & 0 & 0 & 0 & 0 & -2 \end{pmatrix}$$

$$H' = \begin{pmatrix} 0 & 0 & 0 & 0 & 0 & 0 & 0 & 0 \\ 0 & 0 & 0 & 0 & 0 & 0 & 0 & 0 \\ 0 & 0 & -P+Q & -S^* & R & 0 & \sqrt{3/2}S & -\sqrt{2}Q \\ 0 & 0 & -S & -P-Q & 0 & R & -\sqrt{2}R & \sqrt{1/2}S \\ 0 & 0 & R^* & 0 & -P-Q & 0 & \sqrt{1/2}S^* & \sqrt{2}R^* \\ 0 & 0 & 0 & R^* & 0 & -P+Q & \sqrt{2}Q & \sqrt{3/2}S^* \\ 0 & 0 & \sqrt{3/2}S^* & -\sqrt{2}R^* & \sqrt{1/2}S & \sqrt{2}Q & -P & 0 \\ 0 & 0 & -\sqrt{2}Q & \sqrt{1/2}S^* & \sqrt{2}R & \sqrt{3/2}S & 0 & -P \end{pmatrix}$$

where: $\varepsilon = \frac{\hbar^2 k^2}{2m_0}, U = \frac{P_0}{\sqrt{3}}k_z, V = \frac{P_0}{\sqrt{6}}(k_x + ik_z), P = \frac{\hbar^2}{2m_0}\gamma_1\left(k_x^2 + k_y^2 + k_z^2\right),$

$Q = \frac{\hbar^2}{2m_0}\gamma_2\left(k_x^2 + k_y^2 - 2k_z^2\right), \quad S = 2\sqrt{3}\frac{\hbar^2}{2m_0}\gamma_3\left(k_x k_z - ik_y k_z\right),$

$R = -\sqrt{3}\frac{\hbar^2}{2m_0}\left(\gamma_2\left(k_x^2 - k_y^2\right) - 2i\gamma_3 k_x k_y\right), \gamma_1 = -\frac{2}{3}\frac{m_0}{\hbar^2}\left(L' + 2M\right) - 1, \gamma_2 = -\frac{1}{3}\frac{m_0}{\hbar^2}\left(L' - M\right),$

$\gamma_3 = -\frac{1}{3}\frac{m_0}{\hbar^2}N'.$

where $\Delta$ is spin–orbit splitting and $\gamma_1, \gamma_2, \gamma_3$ is Luttinger parameters (*Luttinger & Kohn, 1955*). In the case of GaAs $\gamma_1 = 6.98$ $\gamma_2 = 2.06$ $\gamma_3 = 2.93$ (*Vurgaftman, Meyer & Ram-Mohan, 2001*). In constant media, Luttinger parameters do not depend on coordinates. In the GaAs media case, effective masses of light and heavy holes depend on the direction of hole propagation. In case of Al$_x$Ga$_{1-x}$As Luttinger parameters are chosen as a linear interpolation between the GaAs and AlAs. In AlAs, Luttinger parameters $\gamma_1 = 3.76$, $\gamma_2 = 0.82$, $\gamma_3 = 1.42$.

The Luttinger Hamiltonian we discussed so far is written in the x, y, and z directions corresponding to the principal crystallographic directions [hhe]. We consider structures with modulated potential along the z [111] direction. Thus, the Hamiltonian needs to be rewritten to a new basis. Since SO and CB bands are far from LH and HH for simplicity, and we decouple them. Thus, the 3D Schrodinger equation with effective mass for conduction band electrons can be solved separately from Luttinger Hamiltonian for VB. The Luttinger Hamiltonian, in a new basis, looks like (*Fishman, 1995*):

$$H = \begin{pmatrix} D_{HH} & -S & R & 0 \\ -S^* & D_{LH} & 0 & R \\ R^* & 0 & D_{LH} & S \\ 0 & R^* & S^* & D_{HH} \end{pmatrix}$$

where

$$D_{HH} = -\frac{\hbar}{2m_e}\left[\frac{\partial}{\partial x}(\gamma_1+\gamma_3)\frac{\partial}{\partial x} + \frac{\partial}{\partial y}(\gamma_1+\gamma_3)\frac{\partial}{\partial y} + \frac{\partial}{\partial z}(\gamma_1-2\gamma_3)\frac{\partial}{\partial z}\right] + V$$

$$D_{LH} = -\frac{\hbar}{2m_e}\left[\frac{\partial}{\partial x}(\gamma_1-\gamma_3)\frac{\partial}{\partial x} + \frac{\partial}{\partial y}(\gamma_1-\gamma_3)\frac{\partial}{\partial y} + \frac{\partial}{\partial z}(\gamma_1+2\gamma_3)\frac{\partial}{\partial z}\right] + V$$

$$R = \frac{\hbar}{2m_e}\sqrt{3}\left[\frac{\partial}{\partial x}\left(\frac{2\gamma_3+\gamma_2}{3}\right)\frac{\partial}{\partial x} - \frac{\partial}{\partial y}\left(\frac{2\gamma_3+\gamma_2}{3}\right)\frac{\partial}{\partial y} - i\frac{\partial}{\partial x}\left(\frac{2\gamma_3+\gamma_2}{3}\right)\frac{\partial}{\partial y}\right.$$
$$\left. -i\frac{\partial}{\partial y}\left(\frac{2\gamma_3+\gamma_2}{3}\right)\frac{\partial}{\partial x}\right] +$$
$$+\frac{\hbar}{m_e}\frac{2}{\sqrt{6}}\left[\frac{\partial}{\partial x}(\gamma_3-\gamma_2)\frac{\partial}{\partial z} + \frac{\partial}{\partial z}(\gamma_3-\gamma_2)\frac{\partial}{\partial x} + i\frac{\partial}{\partial y}(\gamma_3-\gamma_2)\frac{\partial}{\partial z} + i\frac{\partial}{\partial z}(\gamma_3-\gamma_2)\frac{\partial}{\partial y}\right]$$

$$S = -\frac{\hbar}{2m_e}\sqrt{3}\left[\frac{\partial}{\partial x}\left(\frac{\gamma_3+2\gamma_2}{3}\right)\frac{\partial}{\partial z} + \frac{\partial}{\partial z}\left(\frac{\gamma_3+2\gamma_2}{3}\right)\frac{\partial}{\partial x} - i\frac{\partial}{\partial y}\left(\frac{\gamma_3+2\gamma_2}{3}\right)\frac{\partial}{\partial z}\right.$$
$$\left. -i\frac{\partial}{\partial z}\left(\frac{\gamma_3+2\gamma_2}{3}\right)\frac{\partial}{\partial y}\right] -$$
$$-\frac{\hbar}{2m_e}\sqrt{\frac{2}{3}}\left[\frac{\partial}{\partial x}(\gamma_3-\gamma_2)\frac{\partial}{\partial x} - \frac{\partial}{\partial y}(\gamma_3-\gamma_2)\frac{\partial}{\partial y} + i\frac{\partial}{\partial x}(\gamma_3-\gamma_2)\frac{\partial}{\partial y} + i\frac{\partial}{\partial y}(\gamma_3-\gamma_2)\frac{\partial}{\partial x}\right]$$

### Funding
The authors received no funding for this work.

### Competing Interests
The authors declare there are no competing interests.

### Author Contributions
- Mikhail Lazarev conceived and designed the experiments, performed the experiments, analyzed the data, performed the computation work, prepared figures and/or tables, authored or reviewed drafts of the article, and approved the final draft.

### Data Availability
The model is available at GitHub and Zenodo:

- https://github.com/Mikelazarev/Low-dimensional-band-structure-model-

- Lazarev Mikhail, & Karlsson Fredrik. (2022). Low-dimensional semiconductor band structure model based on 4-band approximation k.p theory (1.0). Zenodo. Available at https://doi.org/10.5281/zenodo.7438050

### Supplemental Information
Supplemental information for this article can be found online at http://dx.doi.org/10.7717/peerj-matsci.32#supplemental-information.

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
