# Peer review of "Numerical study of valence band states evolution in $Al_xGa_{1-x}As$ [111] QDs systems"

_PeerJ Materials Science, doi:10.7717/peerj-matsci.32_

## Round 0.1 · original submission · Major Revisions

The manuscript requires major revisions and the English language needs to be deeply revised.

Reviewer 1 ·

Basic reporting

The English language needs to be deeply revised. Grammar errors, misspellings, missing articles, broken sentences are very frequent and make the paper difficult to follow.
The extensive literature in the field is not properly referenced. There are many papers on electron/hole complexes in quantum dots calculated at different levels of approximation. The manuscript starts with very general, useless statements on decades of development (which is best found in many review papers), but do not make use use of relevant literature.
In addition to the above comments, the structure of the paper is not well conceived. It is traditional to have a paragraph which terminates the introduction to summarizes what is done and what can be found in the manuscript (often introduced with ‘In this paper..’) which is missing here. Figures are redundant: Fig. 1 reports a number of structures (with undefined labelling) which are NOT investigated. For the two investigated structure, the geometry is repeated in Fig. 2 and 5 (as well as in insets of other figures, so that some of the figures are redundant and the overall results is very confusing.

Experimental design

The results are quite expected. The possible interesting point is the evolution of the polarized absorption with an electric field in regimes of high sensitivity. However, evolution with the field is not systematically investigated.
From the methodological point of view, it is not clearly stated how the space-dependent parameters are treated, and how the Hamiltonian is represented. It seems it is a real space representation, but no detail is given. Boundary conditions, grid parameters ect are not specified. Substantial use of a code described elsewhere is probably made, but self-containedness of the paper is definitely insufficient to evaluate the quality of the results.

Validity of the findings

Results are likely to be valid, that is the correct evalutation of the assumed Hamiltonian, which is justified for the present systems. However, results are expressed in misleading terms. For example, it is stated in the introduction that quantum wells and wires electron-hole recombination occurs via heavy or light holes (both supporting citations are not properly formatted and one of the two impossible to retrieve). I suspect this is only due to the fact that in wells and wires the angular momentum quantization axis is conveniently taken in different directions (growth axis vs free axis) with no physical content. Also, states are often refenced to as HH and LH, while they are clearly a mixture of the two (as the author states but does not consistently use in the description). An obvious consequence of this is that the evolution of the polarization absorption with, e.g., structural parameters is continuous. For example, the optical anisotropy is not ‘switching’ with any parameters, but continuously evolves. Even when crossing of levels occurs, multi-peak structures appear as an expected results of valence band mixing (I also note that the acronym DOLP is not defined), so no clear switching occurs.

Reviewer 2 ·

Basic reporting

1. In the formula of the Hamiltonian matrices, from line 69 to line 96. Many formula should have h_bar instead of h. h_bar = h/(2*pi). You can see them highlighted in the annotated PDF attached. Therefore, please doublecheck your formula and correct those mistakes.
2. Your figure's description does not match your actual figure. In Figure 6, your description for 6(a) does not match with your figure at all. There is no inset in the schematic and your y value is not calculated confinement energy.
3. There are quite some grammatical mistakes and misspelling throughout the manuscript. For example, in line 78, it should be "resulting" instead of "resalting". Line 86-87, "Because transition rate is ..... vector". This sentence does not have subject and verb. Line 127, should be "wavefunction" instead of "wawefunction". In the x axes' label in figure 3, it should be "QD thickness" instead of "QD thikness". And there are other mistakes marked in red in the annotated PDF. Please correct them.
4. In Figure 7(c), there is an orange line. What does it denote? If it refers to HH portion, then it should be pink.
5. In some of your equations, their resolution is lower than others and font size are also different, such as the Hamiltonian matrices on line 81, line 93, line 110-111. Please keep consistency on resolution and fount size across all your equations. It might be helpful to use LaTex to write those equations.

Experimental design

1. When you simulated the effect of an electric field on QD, you chose to fix the QD thickness at 14 nm (line 200). Can you explain why you choose to fix at 14nm specifically?
2. You commented that the high magnetic field for polarization switching requires high field intensities and are not feasible for applications. Therefore, you calculated the spectral change when an electric field is applied. You simulated the optical spectra under E=20000 V/cm (Figure 4) and 9000 V/cm (Figure 9). Can you comment on the feasibility of such intensity of electrical field in applications?
3. You generated DOLP spectra in your results. Does it stand for "degree of linear polarization"? It might not be obvious to the general readers. It will be better to mention the full name of DOLP when you first introduced it and use a few sentences to explain how to read a DOLP spectra.

Validity of the findings

1. You have found that the introduction of electric field can change the polarization properties of QD. It is an interesting finding and might have indications for guiding the development for future polarization switching devices. It will be nice to see how often these computational calculations match with the material's experimental data. In the literature, have people ever reported applying electric field on QD?Are the data usually consistent between computational calculation and experimental data? Are there ways to benchmark the calculation by using some existing experimental results? You can strengthen your importance of finding by answering the above questions.

Annotated reviews are not available for download in order to protect the identity of reviewers who chose to remain anonymous.

Reviewer 3 ·

Basic reporting

This manuscript reports a numerical study of the (“close to valence band edge”) electronic structure and of related optical properties for a particular class of nano-structures. The code, co-developed by the author, is based on a dedicated version of the Luttunger-Kohn (L-H) model, calibrated for the materials of interest. We notice that the L-H (k.p) approach is a broadly applied method for the particular scientific scopes of this manuscript. The code and the related results could be of the interest for the optimization of the features of real fabricated structures as a function of the material composition and geometry parameters. Therefore my general appraisal is that the technical content of the paper could be, in principle, published. However, the major issues must be properly considered in a revised version before the acceptance.

1. The code is available in the github repository and raw data are provided; but numerical results must be validated before the publication. This means that the code should substantiated by the benchmark of the extracted numerical results against literature data or other stable sources, also for simplified configurations (e.g. quantum wells). This is a fundamental aspect to support the reliability of the solutions.

All the aspect of the presentation should be improved. a) Several typos are present or colloquial form of the English language (see e.g. lines 87, 112, 177); b) the literature reference are difficult to find or not properly cited. c) the abstract and the introductions are not effective to present and motivate the method and the results. d) figures are not of the sufficient qualities and contain several typos or not defined acronyms.

Experimental design

A linear interpolation of the model parameters is assumed to consider the alloy effect in the band structure. This is a weak point of the derivation since alloy atoms’ distribution introduces disorder which has an effect in the electronic structure beyond the average between the two parent crystals. This point merits a strong reconsideration and supporting evidences.

The section 2 is devoted to the presentation from the scratch of the derivation of the L-H approximation which can be found in textbook and it be reduced to focus on the main aspect of the method. Moreover, no mention is reported on the implemented numerical techniques expected in this section.

Validity of the findings

The results are presented without any supporting arguments on the numerical values plotted in the figures. For exampled the confinement related energies varies on a more 10eV range (!) as a function of the QD thickness or the Al content (fig 3 a-b) (fig 6 a-b). I would strongly suggest to discuss with at least some qualitative arguments the results.

Reviewer 4 ·

Basic reporting

The author investigated the effect of nanostructure geometry and composition on the optical properties and Valence Band character of Quantum Dots (QDs) and Quantum Dot Molecules (QDMs) using numerical simulations with a Luttinger-Kohn model. They examine GaAs/AlGaAs [111] systems in inverted pyramids and explore optical properties under an external electric field. The author demonstrated a dynamic way to control the optical polarization properties and their evolution, which is important for future nano-optical devices and quantum information technologies. The paper provides valuable information about the potential applications of QDs and the significance of controlling their optical properties. Overall, the paper is well-written and effectively communicates the main findings of the study.

Experimental design

The experimental design describes the methods used in a model, which involves the mean-field (Hartree-Fock) approximation to obtain a Hamiltonian for single electrons, considering the lattice periodicity and using Bloch waves. The model works with the Hamiltonian and solves it in the 4-bands (S and P orbitals) approximation. To take into account an infinite number of other bands, Lowdin perturbation theory was implied. The resulting Hamiltonian in matrix form includes the second-order interactions due to Lowdin renormalization involving states outside the S, P subspace and the spin-orbit splitting. The basis of the Hamiltonian is then changed to the basis of eigenfunctions of momentum operator (HH and LH basis) for optical studies.

Validity of the findings

The results presented in the paper appear to be valid. The author provided clear and detailed information about the methods used to calculate the confinement energies and the parameters used in the simulations. The results are presented in the form of graphs and tables, making it easy to understand the data. The study findings are supported by the calculations of the optical spectra and band mixing characteristic. The author also compared their results with previous research and found their results consistent with earlier research. However, since the study is based on simulations, further experimental validation of the findings is necessary to confirm their accuracy.

Additional comments

No further comments

---

## Round 0.2 · Minor Revisions

The manuscript is a sound approach to calculation of important variables for these materials. However, please look at the specific comments of the reviews and correct these for a resubmission of your work. It is critical to also insure the manuscript has been edited for proper language and grammar throughout. If there are issues in this regard PeerJ offers help along these lines. We appreciate you choosing PeerJ for consideration of your work.

**Language Note:** The Academic Editor has identified that the English language must be improved. PeerJ can provide language editing services - please contact us at [email protected] for pricing (be sure to provide your manuscript number and title). Alternatively, you should make your own arrangements to improve the language quality and provide details in your response letter. – PeerJ Staff

Reviewer 1 ·

Basic reporting

No comment

Experimental design

No comment

Validity of the findings

No comment

Additional comments

The resubmitted version of the paper takes into consideration part of the criticisms from previous referees.
There are still several presentation issues.
1 first, there is a disturbing number of grammar errors for a revised version. It is a sign a respect for the referees and the readers to carefully cross-check the manuscript. I believe it is not the task of the referee to provide a list of errors, which would take much to long here (and start from the abstract and the 1st paragraph).
2 In the abstract it is stated that the source code is presented. This is not true. The code in itself is not discussed, nor its use, in any sense (this would not be the right review to do it, indeed)
3 In the introduction it is stated that the recombination is via LHs in QWR rather then HH in QWs. I do not understand where the author got this idea. in Vouilloz et al is is only shown that WHEN the excitation occurs via LH (which is never the fundamental state) polarization changes to to HH-LH mixing. This also shown plainly in App Phys Lett 69, 2965 (1996) (not referenced here).
4 lines 112-119 present textbook material
5 line 137. What does it mean identical polarization dependence? identical between what and what?
6 There is a LARGE number of citations which are incomplete (i.e. they do not contain Volume, page, year)

Reviewer 2 ·

Basic reporting

The author addressed most of the provided feedback; however, a few issues remain unresolved:
1.1 Despite being correctly displayed in the Word document, the symbol "h_bar" was still inaccurately represented as "h" in the PDF proof. Please rectify.
1.2 Furthermore, the issue related to the orange line in Figure 7(c) has not yet been resolved and still requires attention.

Experimental design

The author responded to all my inquiries concerning experimental design. However, while addressing the initial point, a grammatical error emerged as the author inadvertently used the term "exited" instead of "excited" when referencing the excited state. This same grammatical error recurred in line 322 as "exited CB."

Validity of the findings

The author made a reference to their prior publication to illustrate the alignment between their computational findings and experimental results. To enhance the manuscript, it would be beneficial for the author to explicitly include this point and additionally explore its relevance in other studies in the literature.

---

## Round 0.3 · Minor Revisions

Thank you for submitting your revised manuscript. Please look over the comments made by the reviewer with specific changes that must be addressed in the manuscript as recently submitted.

Reviewer 1 ·

Basic reporting

The author has partially met previous comments from the referees. The paper is now new to publication standards, but the author is urged to consider the following:
- There is still a large number of misprints/uncorrect sentences, e.g.
l. 74 'it is' -> 'For example'
l.197 'take place' -> 'takes place'
l.201 'aria' ?
- the acronym QD is defined twice
- l. 87, 'little attention was paid to the related VB structure'. If this refers to the specific pyramidal structures, this might be true. In general, however, the VB of single and double quantum dots has been the subject of extensive investigations within the k.p and other theoretical and experimental methods
- equations are unnumbered until final equations (1), (2)
- this, for example, makes it difficult to refer to equation at line 116, which is ambiguous, since some of the terms are scalar, some others are matrices. Identity matrices may help here.
- l. 119, it is difficult to imagine 'an infinite number' of bands to be taken into account. In k.p specific bands are taken into account by the folding Lowdin method
- line 133, there is a strange formatting of the matrix element
- in several places the term 'dynamic' is used in relation to the electric field. It is not clear what 'dynamic' refers to. There is not dynamical modulation of the field here. Calculations are performed at selected fixed fields and no dynamics is involved. Tunability, which I suspect is what the author means, has nothing 'dynamic' from my point of view.
- please use scientific notation throughout: 30000 V/cm is different from 30 x 10^3 V/cm or 30 kV/cm
- there is no comparison to any experimental data whatsoever. The author states that such structures have been realized, but either there are not data (which should be stated clearly, to motivate experimental work) or the author should make an effort to compare his calculations to available evidence.
- there are still unproperly formatted citations missing complete journal reference

Experimental design

no comment

Validity of the findings

no comment

Additional comments

no comment

---

## Round 0.4 · accepted · Accept

As you have addressed the concerns of the previous reviewer the manuscript is accepted and proofs will be sent shortly by the editorial office.